# Effect of Addition of Fermented Soy Sauce on Quality Characteristics of Pork Patties during Refrigerated Storage

**DOI:** 10.3390/foods11071004

**Published:** 2022-03-29

**Authors:** Yun-a Kim, Sanghun Park, Yunhwan Park, Gyutae Park, Sehyuk Oh, Jungseok Choi

**Affiliations:** Department of Animal Science, Chungbuk National University, Cheongju 28644, Korea; rladbsdk0621@naver.com (Y.-a.K.); pksaho@chungbuk.ac.kr (S.P.); yhp056@naver.com (Y.P.); qkrrbxo113@naver.com (G.P.); opk1122@naver.com (S.O.)

**Keywords:** fermented soy sauce, storage property, refrigerated storage, antioxidant activity, pork patty, natural oxidant

## Abstract

This study aimed to determine whether fermented soy sauce has a mutually synergistic effect on the quality and storage properties of pork patties, and to investigate the effects on the availability and physicochemical properties of various taste ingredients of soy sauce, a traditional Korean food ingredient. The experimental groups were as follows: Control (−): No additives; Control (+): 0.1% ascorbic acid; T1: 1% fermented soy sauce; T2: 3% fermented soy sauce; T3: 5% fermented soy sauce. No significant difference was detected in moisture, protein, and fat among the various treatment groups; however, ash content and water holding capacity increased and texture properties improved with the concentration of fermented soy sauce. The addition of fermented soy sauce during refrigerated storage for 10 days showed a positive effect on the storage properties. The peroxide value, content of thiobarbituric acid reactive substances and total phenolics, and 2,2-diphenyl-1-picrylhydrazyl free radical scavenging activity differed significantly in pork patties with different treatments and storage intervals. The effect of fermented soy sauce on the overall quality and storage properties of pork patties during refrigerated storage is relatively unknown. These findings demonstrate that the addition of fermented soy sauce improves the quality properties and antioxidant activity of pork patties.

## 1. Introduction

Global meat consumption increased by 500% from 1992 to 2016, and continued growth in meat consumption is expected [1]. The production and consumption of processed meats have increased every year since 2015, as meat consumption has gradually increased worldwide [2]. Meat and processed meat products are susceptible to deterioration in quality as they contain abundant components such as salt, heme iron, and endogenous phospholipids [3]. In addition, meat is one of the most perishable foods because of its high nutritional and water content [4]. The most common form of chemical deterioration is oxidation of meat lipids. Lipid oxidation depends on the meat’s chemical composition, light, oxygen, and storage temperature [5]. Several antioxidants such as nitrate and ascorbic acid have been used as additives or supplements in meat or meat products to prevent lipid oxidation [6,7]. Most claims concerning the idea that processed meat is not good for health come from the use of certain ingredients that are added during processing as well as the processing conditions [8,9]. Several studies have been conducted to replace chemical additives. Pateiro et al. [10] reported that addition of essential oil as an antioxidant in chicken breast products enhanced the oxidative stability of the products. Plant products such as fruits (grapes, pomegranates, date, kinnow), vegetables (potatoes, potato, drumstick, pumpkin), herbs (curry, nettle), and spices (tea, rosemary, oregano, cinnamon, sage, thyme, mint, ginger, clove) have been used to preserve and improve the overall quality of meat and meat products [11,12,13,14,15,16,17,18,19,20,21,22,23]. Soy sauce is a traditional fermented seasoning made from soybeans and it is used as a seasoning for various meat-based dishes in East Asian countries [24,25]. Soy sauce contains soy-derived salt (approximately 15%–20% of salt concentration), water (approximately 50–70%), peptides, isoflavones, as well as sugar-free and organic acids created during fermentation [26,27,28]. Soy sauce is rich in amino acids, peptides, and simple sugars, providing a rich precursor for Maillard reactions. The Maillard reaction plays a large role in preventing oxidation. The various mechanisms involved in the antioxidant activity of Maillard reaction products (MRPs) include radical chain breaking activity [29], scavenging of reactive oxygen species, and a delay in the formation of primary and secondary oxidation products due to hydroperoxide and metal chelate decomposition [30]. Therefore, MRPs formed in soy-cooked products can delay oxidative waste [31]. Therefore, adding soy sauce to patties will affect antioxidant capacity through Maillard reaction after cooking. In addition, it has been reported that soy sauce is well known for having a strong antioxidant function in some foods, including meat products [32,33]. The antioxidant effect of soy sauce has been investigated and verified in beef patties [34]. Moon and Cheigh [35] proved that the soy sauce of cooked beef plays an important role as an antioxidant. Fermented soy sauce is attracting attention as a functional material beyond its use as a traditional seasoning for flavor. However, studies on the overall quality and storage properties of pork patties supplemented with fermented soy sauce during refrigeration storage are rare.

In this study, fermented soy sauce at different concentrations was added to pork patties in order to confirm the availability of the sensory properties of various taste ingredients in soy sauce. The sauce was expected to have a mutually synergistic effect on the quality and storage properties of pork patties.

## 2. Materials and Methods

### 2.1. Pork Patties

#### 2.1.1. Preparing Pork Patties

Ground pork hind legs were purchased, and blood was removed. Purified salt, ascorbic acid, and fermented soy sauce (Korean soup soy sauce, Chungjungwon) were mixed with ice so that the temperature of the mixture was maintained at 10 °C for 5 min using a mixer. Fermented soy sauce consisting of 3470 mg of sodium, 5.9 g of carbohydrate, 5.3 g of sugar, 5.3 g of protein, and 0 g of fat per liter was used. The second grinding was performed using a grinder. Patties (90 g each) to be used for all treatments were prepared in a circular shape. The composition ratio of each treatment is shown in Table 1. The prepared patties were wrapped without heating and stored in a refrigerator at 4 °C. Three refrigerated pork patties from each treatment group were analyzed according to the storage period (0, 5, and 10 days). The experiment was repeated three times.

#### 2.1.2. Sodium Content of Pork Patties

The sodium content of soy sauce and salt is as follows; in the case of salt, the sodium content is 387.58 mg (per 1 g of salt), and in the case of soy sauce, the sodium content is 54.32 mg (per 1 g of soy sauce). Therefore, the sodium content contained in the patty (100 g) of the control (−) and control (+) sample is 620.13 mg in total (0.62% of the patty). The sodium content contained in the patty (100 g) of the T1, T2, T3 samples is 674.45, 728.77, and 783.09 mg in total (0.67%, 0.72%, and 0.78% of the patty).

### 2.2. Physicochemical Properties

#### 2.2.1. Moisture Content

Moisture contents were measured in accordance with the AOAC method [36]. The aluminum plates were dried in a dry oven for 30 min before the experiment. The weight of the dried aluminum plate was measured. Samples of approximately 1 g were taken from pork patties from all treatment groups. The samples were placed on an aluminum plate and their initial weight was determined and recorded. It was dried in a dry oven at 105 °C for 16 h. Later, the samples were quickly put into a desiccator to minimize the change in moisture in all treatments. After putting the samples in a desiccator and allowing them to cool for approximately 30 min, the moisture content in the samples was calculated as follows.
$$\mathrm{Moisture}=\frac{\mathrm{weight}\;\mathrm{of}\;\mathrm{alumium}\;\mathrm{plate}-\mathrm{weight}\;\mathrm{after}\;\mathrm{drying}}{\mathrm{sample}\;\mathrm{weight}}\times 100$$

#### 2.2.2. Crude Ash

Crude ash contents were measured in accordance with the AOAC method [36]. Approximately 1 g of a sample was put in a crucible and weighed, and the crucible number and weight were recorded. After drying the sample in a drying oven for 2–3 h, the dried sample was incinerated at 540 °C for 10 h in an incinerator. Next, the sample was transferred to a desiccator using forceps, allowed to cool for approximately 1 h, and then weighed. The ash content was calculated as follows.
$$\mathrm{Ash}\;\mathrm{content}\left(\%\right)=\frac{\mathrm{weight}\;\mathrm{after}\;\mathrm{incinerating}-\mathrm{weight}\;\mathrm{before}\;\mathrm{incinerating}}{\mathrm{sample}\;\mathrm{weight}}\times 100$$

#### 2.2.3. Crude Fat

The crude fat content was determined using the procedure established by Folch et al. [37]. The freeze-dried sample (0.5 g) was put in a conical tube. Next, 20 mL of Folch solution was added to the tube and the contents were shaken well. After making up the volume to 5 mL using Folch solution, the tube was placed in a refrigerator at 4 °C for 24 h. After filtering the solution with filter paper, 5 mL of Folch solution was added to the tube through the filter paper. Distilled water (10 mL) was added to the tube such that any contents remaining on the wall of the tube were removed. After centrifuging the mixture for 20 min at 100× *g* at 21–23 °C, the supernatant was removed. The liquid from the lower layer was transferred to an oven-dried and weighed beaker. After drying the chloroform overnight in a hood and drying in a drying oven for approximately 30 min, the weight of the beaker was measured. The fat content was calculated as follows:$$\mathrm{Wet}\;\mathrm{sample}\;\mathrm{weight}=\frac{100\times \mathrm{sample}\;\mathrm{weight}}{100-\mathrm{moisture}\;\mathrm{content}}$$
$$\mathrm{Fat}\;\mathrm{content}=\frac{\mathrm{beaker}\;\mathrm{weight}\;\mathrm{after}\;\mathrm{drying}-\mathrm{beaker}\;\mathrm{weight}}{\mathrm{wet}\;\mathrm{sample}\;\mathrm{weight}}\times 100$$

#### 2.2.4. pH

Distilled water was added to 10 g of pork patty samples to measure the pH. All samples were homogenized for 30 s using a homogenizer (Bihon Seiki, Ace, Osaka, Japan), followed by measurement of pH using a pH meter (Mteeler Delta 340, Mettler-Tolede, Ltd., Cambridge, UK).

### 2.3. Water Holding Capacity (WHC)

WHC was determined using a method previously described by Laakkonen et al. [38]. A 2 mL centrifuge tube with fine holes was weighed. Crushed sample (0.5 ± 0.05 g) was added into the filter tube at the top of the centrifuge tube and then heated at 540 °C for 20 min. Later, the sample was allowed to cool for 10 min. The upper filter tube was placed in the lower part of the centrifuge tube and centrifuged for 10 min at 45× *g*. WHC was calculated as follows.
$$\mathrm{Free}\;\mathrm{moisture}\;=\left[\frac{\mathrm{Weight}\;\mathrm{before}\;\mathrm{centrifugation}\;-\;\mathrm{Weight}\;\mathrm{after}\;\mathrm{centrifugation}}{\mathrm{Sample}\;\mathrm{weight}}\right]\times \;\mathrm{Fat}\;\mathrm{coefficient}\;\times 100$$
Fat coefficient =(1− fat content)×100
$$\mathrm{WHC}\left(\%\right)=\left[\frac{\mathrm{total}\;\mathrm{moisture}-\mathrm{free}\;\mathrm{moisture}}{\mathrm{total}\;\mathrm{moisture}}\right]\times 100$$

### 2.4. Texture Profile Analysis (TPA)

To measure the texture (hardness, springiness, cohesiveness, chewiness) of the pork patties, they were cut into 1 cm^3^ pieces. Next, testing of TPA was performed using a rheometer (Compac-100, sun Scientific Co., Tokyo, Japan). The program used was RDS (Rheology Data System) ver 2.01. The table speed was 110 mm/min, graph interval was 20 m/s, and load cell (max) was 10 kg.

### 2.5. Sensory Test

A sensory test was performed by five evaluation panelists (2 women and 3 men aged 24–28 years with an average age of 24.8 years) in the Department of Animal Science, Chungbuk National University. The evaluation panelists were trained according to the guidelines of American Science Association [39] and were trained on product descriptions and terminology. All patties were cooked using a pre-heated pan, until the samples’ internal temperature of 72 ± 1 °C was achieved for 7–8 min. Patties were cut into blocks with a thickness of 1.5 cm, length of 1.5 cm, and width of 1.5 cm and were provided on a white plate. The sensory test was conducted at room temperature (18–21 °C). After eating from one sample, evaluation panelists were asked to rinse their mouths with water and eat the next sample after waiting 1–2 min for the evaluation. The evaluation factors consisted of the following six items: color, flavor, juiciness, saltiness, chewiness, and overall preference, and the evaluation was performed using a 5-point rating method. Each item was scored from 1 point (worst appearance, worst flavor, driest, less preferred saltiness, toughest, worst overall acceptance) to 5 points (best appearance, most flavor, driest, preferred saltiness, toughest, best overall acceptance).

### 2.6. Meat Color

The surface meat color of the pork patties was measured with a Spectro Colorimeter (Model JX-777, Color Techno, System Co., Tokyo, Japan). A white fluorescent lamp (D65) was used to represent the L, a, and b values of the Hunter Lab color system (L = brightness; a = redness; b = yellowness).

### 2.7. Storage Properties Analysis

#### 2.7.1. Total Microbial Count (TMC)

Total microbial count (TMC) was determined using the method described by Aliakbarlu et al. [40]. The total bacterial count was determined using a total plate count agar (Difco plate count agar, BD, New Jersey, USA). Using a stomacher (400 Circulator, Seward Ltd., Worthing, UK), 10 g of homogenized ground meat was mixed with 90 mL of sterile distilled water. Thereafter, 1 mL of the mixture was transferred to sterile distilled water using a pipette and diluted 10-fold. If necessary, the dilution factor was increased and used in the experiment. One milliliter of each diluted solution was inoculated into the medium and cultured at 37 °C for approximately 2 days. The number of colonies generated after cultivation was measured, and the result was expressed as log 10 colony forming unit (CFU)/g.

#### 2.7.2. 2-Thiobarbituric Acid (TBA)

2-Thiobarbituric acid (TBA) value was measured using the modified extraction method of Witte et al. [41]. The sample (10 g) was homogenized with 15 mL of cold 10% PCA solution and 25 mL of distilled water using a homogenizer at 1100 g for 15 s. After homogenization, the whole eluate was transferred to Whatman No. 2 filter paper using ø150 mm filter paper. Using a pipette, 5 mL each of filtrate and TBA solution was transferred into the numbered tube and the lid was closed. After mixing well using a vortex mixer, 5 mL each of distilled water and 0.02 M TBA solution was mixed and used as a blank test tool. After mixing, the surface was sealed with Para film and the tube was placed in a tube rack. After incubation for 16 h in a cool, dark place, absorbance was measured at 529 nm using a spectrophotometer (Model JX-777, Color Techno. System Co., Tokyo, Japan).

#### 2.7.3. Determination of 2,2-Diphenyl-1-Picrylhydrazyl (DPPH) Free Radical Scavenging Activity

The antioxidant activity of the sample was determined using a modified DPPH free radical scavenging assay [42]. A total of 5 g of each patty and 45 mL of methyl alcohol (SAMCHUN Pure Chemical Co. Ltd., Pyeongtaek, Korea, 99.5%) were homogenized for 1 min with a homogenizer (Bihon seiki). The solution was filtered using 150 mm filter paper (Advantec, Tokyo, Japan) to remove impurities, and then centrifuged at 12,000× *g* for 30 min with a centrifuge (5424R, Eppendorf, Hamburg, Germany). Solutions excluding precipitates were used. Thereafter, samples, blanks, and references were prepared as follows: Sample: 2 mL of solution, 1 mL of 99% DPPH, 2 mL of methyl alcohol; Blank: 5 mL of methyl alcohol; Reference: 1 mL of DPPH, 4 mL of methyl alcohol. After blocking the light, the samples were left at room temperature in a dark room for 20 min. The absorbance of the solution was measured at 517 nm using a microplate spectrophotometer (Microdigital Co., Ltd., Seongnam, Korea). The scavenging activity of the patty sample against DPPH radical was calculated as:$$\mathrm{DPPH}\;\mathrm{radical}\;\mathrm{scavenging}\;\mathrm{activity}\left(\%\right)=\left[1-\left(\frac{\mathrm{sample}\;\mathrm{of}\;\mathrm{absorbance}}{\mathrm{reference}\;\mathrm{of}\;\mathrm{absorbance}}-\;\mathrm{blank}\;\mathrm{of}\;\mathrm{absorbance}\right)\right]\times 100$$

#### 2.7.4. Estimation of Peroxide Value (POV)

The lipids were extracted from the 5 g of patty with 25 mL of solvent mixture (acetic acid:chloroform mixture, 3:2). The POVs of the extracts were determined using the modified method of Lea [43]. The mixture was shaken thoroughly, and 1 mL of saturated potassium iodide solution was added. Then, it was kept in the dark at room temperature for 10 min. After stabilization, 30 mL of distilled water and 1 mL of starch solution (1 g/100 mL) were added to the solution and titrated with 0.01 N Na_2_S_2_O_3_ until it was colorless. POVs were calculated as follows:$$\mathrm{POV}\left(\frac{\mathrm{meq}}{\mathrm{kg}}\right)=\left(S-B\right)\times \frac{F\;\mathrm{mol}\;\mathrm{equiv}}{L\left(N\right)}\times \frac{1000}{W}$$

S = titration amount of sample; B = titration amount of blank; F = titer of 0.01 N Na_2_S_2_O_3_; *N* = normality of Na_2_S_2_O_3_; *W* = sample weight (g).

The results are expressed as milliequivalent peroxide O_2_/kg meat.

### 2.8. Organic Acid Analysis by High Performance Liquid Chromatography (HPLC)

HPLC analysis was performed using LC/Q-TOF (Bruker Biosciences, Billerica, MA, USA, Maxis 4G, 20 Hz). For the mobile phase, 5 mM H_2_SO_4_ was used. The flow rate was 0.4 mL/min. The chromatogram was observed at 210 nm and the injection volume was 5 µL. The column utilized was a C18-column (Synergi 2.5 μm, Hydro-RP, 100 Å, 100 mm × 2 mm). Citric, malonic, lactic, acetic, oxalic, and fumaric acids were obtained from SAMCHUN Pure Chemical Co. Ltd. (Pyeongtaek, Gyeonggido, South Korea).

The patties (total weight 10 g) were homogenized with 0.01 N H_2_SO_4_ using a homogenizer (Bihon seiki) before incubation in a 30 °C water bath for 30 min and then filtered. For the first filtration, 150 mm filter paper (Advantec) was used, and for the second filtration, a 40 µm membrane (SPL Life Science, Pocheon, South Korea) was used. The filtrate was centrifuged at 21,000× *g* for 10 min in a centrifuge (5424R, Eppendorf, Hamburg, Germany). After that, the supernatant was taken and filtered through a 0.2 μm membrane (Corning, NY, USA), and the filtrate was injected into the HPLC column.

### 2.9. Statistical Analysis

All experiments were repeated at least three times. Statistical analysis was performed through the General Linear Model procedure of the SAS program (Statistics Analytical System, Cary, NC, USA, 1999). The significance (*p* < 0.05) was determined using Duncan’s multiple test for comparing the means of the treatment groups.

## 3. Results and Discussion

### 3.1. Sodium Content of Pork Patties

Excessive sodium intake has a direct relationship between cardiovascular disease and subsequent risk of stroke and is unhealthy [44]. However, considering the salt content of the soy sauce, it was judged that the addition of the soy sauce does not significantly affect the sodium content of the patty. Therefore, it was judged that the increase in sodium content due to the addition of soy sauce is not significant.

### 3.2. Physicochemical Properties of Pork Patties Supplemented with Soy Sauce

Table 2 shows the physicochemical properties of pork patties supplemented with fermented soy sauce. There was no significant difference between the control (−) and any treatment groups with respect to the moisture, protein, and fat. Regarding the ash content, the patties treated with 3% and 5% fermented soy sauce showed a significant difference (*p* < 0.05) from the control (−) and other treatment groups. According to the Food Ingredients Table, the physicochemical properties of pork patties accounted for moisture content 70.4%, ash content 16.7%, crude protein 7.7%, and crude fat 0.3%, based on the general ingredient specifications of traditional soy sauce. Thus, fermented products seem to have influenced the physicochemical properties [28]. Regarding pH, treatment with ascorbic acid alone showed a significant difference (*p* < 0.05), with the pH lower than that in the control (−) and other treatment groups. Ascorbic acid has a pH of 2.5. It has been reported that the pH of beef patties reduces upon the addition of 1% ascorbic acid to the patties [45]. Accordingly, it was considered that the significant difference (*p* < 0.05) observed in the treatment group to which only ascorbic acid was added was a result of the low pH of the ascorbic acid.

### 3.3. WHC of Pork Patties Supplemented with Soy Sauce

Table 3 shows the WHC of pork patties supplemented with fermented soy sauce. The WHC showed the lowest results in control (+) treated with ascorbic acid. This is seen as a result of low pH. It is consistent with the results of Honikel et al. [46] that the WHC increases as the pH value moves away from the isoelectric point pH of the protein. In control (+), the WHC tended to be significantly lower according to the tendency of pH. On the other hand, the WHC of the treatments with added soy sauce tended to be significantly higher.

### 3.4. Texture Properties of Pork Patty Supplemented with Soy Sauce

Table 4 shows the texture properties of pork patties supplemented with fermented soy sauce. Springiness and cohesiveness results in all treatment groups indicated that neither the control (−) nor all the treatment groups showed a high grade or a significant difference (*p* < 0.05). The hardness of patties in groups T2 and T3 treated with 3% and 5% fermented soy sauce, respectively, was significantly higher than that of the patties in the other treatment groups (*p* < 0.05). The chewiness of patties in groups T2 and T3 was significantly higher than that of the patties in the control (−) and other treatment groups (*p* < 0.05). A study by McGough et al. showed similar results, with hardness and chewiness affected by various levels of soy sauce content, but there was no difference in elasticity and cohesion [47]. The texture properties of the patties treated with fermented soy sauce tended to improve with the concentration of soy sauce. Park and Kim [48] reported that the pork emulsions of viscosity increased following treatment with a high salt concentration during the preparation of pork emulsions. Similarly, another study reported that the hardness value increased with salt concentration during preparation of meat products [49,50,51]. In addition, it has been reported that treatment with a high salt concentration during their preparation increases the hardness and shear force of sausages [52,53,54,55]. Jiménez-Colmenero et al. [51] reported that the protein solubility increases with an increase in the amount of salt added. Accordingly, it was considered that the salt component of the soy sauce increased the binding power by promoting the extraction of salt-soluble protein from pork patties, as well as improving the texture of pork emulsions.

### 3.5. Sensory Evaluation of Pork Patties Supplemented with Soy Sauce

Sensory evaluation of pork patties supplemented with fermented soy sauce is shown in Table 5. The addition of fermented soy sauce did not cause any significant difference in color, flavor, juiciness, and chewiness; however, saltiness score was significantly lower in T3. A low saltiness score means that it is not the preferred saltiness. It has been reported that fermented soy sauce contains 15–20% of salt and natural substances that add saltiness, making it possible to prepare low-sodium processed meat products [34]. However, we believe that because the same amount of salt was added to both control (−) and treatment groups, it is judged that the saltiness score decreases as the soy sauce is added.

T3 with 5% fermented soy sauce showed good results in most of the experiments; however, it was considered that when fermented soy sauce was added to the actual food production, T2 with 3% added fermented soy sauce was more appropriate. If the amount of added fermented soy sauce is similar to that added in T3, it will be necessary to adjust the amount of salt.

### 3.6. Meat Color of Pork Patty Supplemented with Soy Sauce

Table 6 shows the results of meat color on the day of supplementation of pork patties with fermented soy sauce. Compared to the control (−), T2 and T3 with added fermented soy sauce showed lower L values (brightness) (*p* < 0.05), but the value of T1 was similar to that of the control (−). Similarly, it has been reported that the L value of fermented sausage decreases with the addition of soy sauce. This is considered to be due to the unique color characteristics of soy sauce [34]. The a value, which represents the redness, did not show a significant difference. The b value indicating the yellowness was high in the fermented soy sauce treatment groups, but there was no significant difference. The high b value could be attributed to melanoidin, a powerful antioxidant [24]. The melanoid reaction is caused by sugars and amino acids known to contribute to the dark brown color formation characteristic of fermented soy sauce.

### 3.7. Storage Properties of Pork Patties Supplemented with Soy Sauce during Refrigeration Storage

The storage properties of pork patties supplemented with fermented soy sauce by storage time are shown in Table 7. The TBA value is primarily used to assess lipid oxidation in meat and meat products. TBA, indicating fatty acid fatness, did not show a significant difference in T3 with 5% fermented soy sauce compared to that of the control (−) to which ascorbic acid was added on Days 0 and 5 of refrigerated storage. At 10 days of refrigerated storage, the TBA values were the highest among all treatments compared to that of the control (−) and the group with 1% soy sauce. However, Control (+) was 0.49 ± 0.15 mg, T2 was 0.88 ± 0.39 mg, and T3 was 0.74 ± 0.35 mg, showing a significant difference (*p* < 0.05). Therefore, the results indicate that 1% of soy sauce does not have an antioxidant effect, while 3% or 5% of soy sauce prevent lipid oxidation similar to the addition of ascorbic acid, a synthetic antioxidant. Soy sauce is rich in amino acids, peptides, and simple sugars, providing a rich precursor for the Maillard reaction. The Maillard reaction in the soy sauce plays a great role in antioxidant protection. According to Wang et al. [24], the Maillard reaction in the soy sauce results in an antioxidant effect due to the hydroxyl and amine groups. Additionally, the soy sauce contains isoflavones and flavonoids, which are antioxidants derived from soy [35]. Moreover, it has been reported that phenolic compounds present in fermented soy sauce occupy most of the natural antioxidant system in the plant world and prevent oxidation of lipids, scavenging free radicals [56]. Accordingly, it was considered that adding 3% or more of soy sauce is an effective method of inhibiting lipid oxidation during refrigerated storage after pork patty production.

In the case of TMC, there was no significant difference in all treatments for 10 days of storage. According to Kargiotou et al. [57], when raw beef was marinated in soy sauce-based marinade, it was effective in inhibition of microbial growth on marinated beef samples. However, in this experiment, it was judged that a small amount of soy sauce was added, so there was no effect of microbial inhibition in the pork patty. Therefore, when less than 5% of soy sauce was added to the pork patty, it did not affect the TMC.

The antioxidant activity (%) of the patty showed significant results within the storage period (*p* < 0.05) (Figure 1). Overall, it tended to decrease statistically as the storage period of patties increased. These results are consistent with the findings of Prommachart et al. [58], who reported that an increase in the number of storage days reduces the radical elimination of beef patties. Patties with ascorbic acid and soy sauce showed a higher antioxidant activity than the control (−). The control (−) showed lower antioxidant activity than the treatments with soy sauce and ascorbic acid. Control (+) with ascorbic acid significantly exhibits the highest antioxidant activity among treatments and tends to be similar to the studies of Kim and Chin [59]. This could be a result of the addition of ascorbic acid, a chemical antioxidant. In addition, the DPPH radical scattering activity of the patty increased with the additional amount of soy sauce. T3 (the treatment at which 3% of soy sauce was added) showed a significantly higher antioxidant activity than t1 and t2 in all storage days. In contrast, t1 treated with 1% of soy sauce showed a significantly lower antioxidant activity than t2 and t3 in all storage days. Previous studies have suggested that soy sauce can act as an antioxidant in food [30,60]. According to Aoshima and Ooshima [32], the soy sauce shows strong DPPH radical-scavenging activity as well as a high polyphenol concentration. Based on these previous studies and our experimental results, it can be explained that the antioxidant activity of the patty increases with the increase in the amount of soy sauce added during patty manufacturing.

The POV of each treatment tended to increase statistically with storage time (Figure 2). These results are consistent with the studies that reported that the POV of pork patties decreases as the storage period increases [59]. As shown in Figure 2, the POV of the control (−) was significantly the highest during the storage period excluding Day 0. On Day 0, no significant difference was detected between processing units, but on Day 5, the ascorbic acid content of patties from the soy sauce treatment was lower than that of the control (−). Thus, it can be explained that the soy sauce acts as an antioxidant, similar to ascorbic acid. In addition, the decrease in POV after Day 10 could be highly related to the mechanism of lipid oxidation [61].

### 3.8. Organic Acid Analysis of Pork Patties Supplemented with Soy Sauce during Refrigeration Storage

HPLC of the patty showed an organic acid profile in which citric acid, malonic acid, acetic acid, oxalic acid, and fumaric acid exist (Figure 3). In the control (−), all remaining compounds (citric acid, malonic acid, acetic acid, oxalic acid) except fumaric acid were found in the patties (Table 8, Figure 4). However, from a quantitative point of view, substantial differences were noticed. Quantification of identified compounds was followed by citric acid as a major compound in all analyzed patties, generally acetic acid and malonic acid. The organic acid of the patty stored for 10 days showed no significant difference or tendency to increase from the control (−) group of the patty stored for 0 days. The oxalic acid content reported in the present study was similar to the findings of a previous study [62]. The organic acid content between patties did not show a significant difference as a whole, but the oxalic acid content was high in patties with soy sauce. It is believed that soy sauce acts as an antioxidant in patties, referring to the study by Kayashima and Katayama [63], that reported that oxalic acid can be used as an antioxidant in food. In addition, oxalate has been reported to have a strong chelating availability [64,65], which is consistent with the results of the previous DPPH experiment.

## 4. Conclusions

Addition of soy sauce to pork patties resulted in a positive effect on texture properties. As the level of soy sauce increased, the salt component of the soy sauce helped to extract the salt-soluble protein from pork patties, increasing the binding and shear forces. In addition, the soy sauce showed a positive effect on the storage properties. Adding 3% or more soy sauce to pork patties resulted in a lower preference for saltiness. Therefore, when 3% or more of soy sauce was added, it was desirable to reduce the amount of salt in the patty to adjust the preference of the saltiness. Addition of 3% or more of soy sauce to pork patties showed a great effect on lipid oxidation compared to the addition of ascorbic acid. In addition, patties with ascorbic acid reduced WHC by reducing pH, while adding soy sauce played a positive role in patties’ WHC. Therefore, soy sauce has the potential to replace chemical antioxidants in the production of commercial pork patties. In terms of storage properties, it was found that adding more than 3% of soy sauce could prevent lipid oxidation. This study was conducted to investigate the characteristics of pork patties supplemented with soy sauce as a natural antioxidant to replace chemical antioxidants used in preparing meat products. The findings of this study demonstrate that soy sauce not only plays a role as a natural antioxidant, but also has a positive effect on texture properties.

## Figures and Tables

**Figure 1 foods-11-01004-f001:**
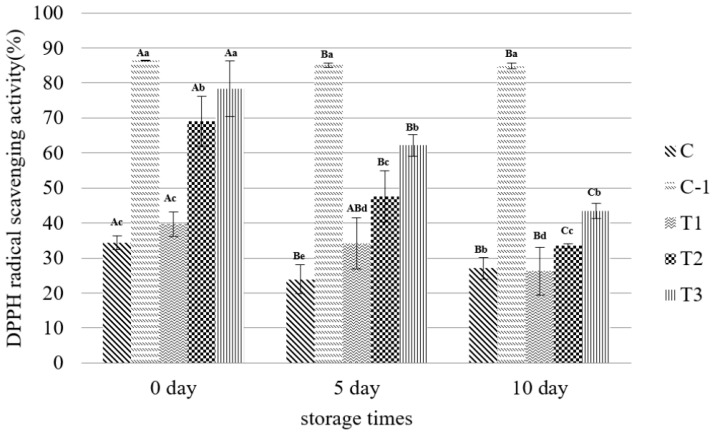
Antioxidant activity (%DPPH) of patties with different levels of soy sauce during 10 days of storage. ^A–C^ Least square means with different letters within the same day of storage time are significantly different (*p* < 0.05). ^a–e^ Least square means with different letters within the same treatments are significantly different (*p* < 0.05).

**Figure 2 foods-11-01004-f002:**
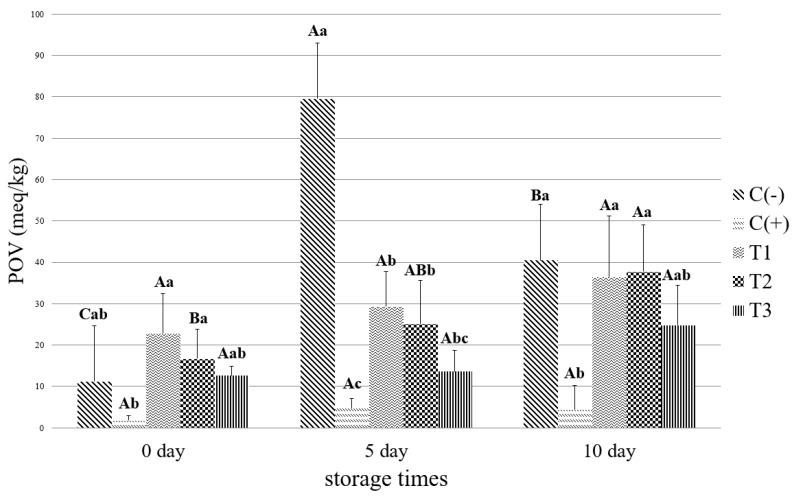
Peroxide value (POV) of patties with different levels of soy sauce during 10 days of storage. ^A–C^ Least square means with different letters within the same day of storage time are significantly different (*p* < 0.05). ^a–c^ Least square means with different letters within the same treatments are significantly different (*p* < 0.05).

**Figure 3 foods-11-01004-f003:**
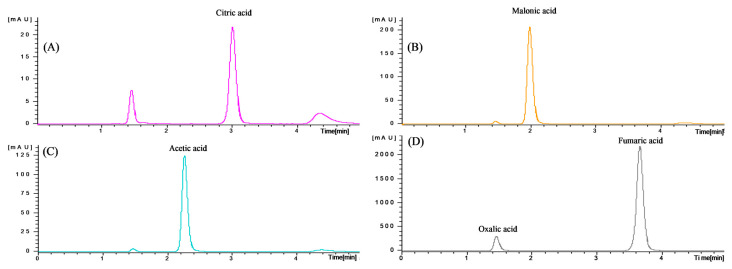
Chromatogram of the standard solution of the organic acids. (**A**) Citric acid, (**B**) malonic acid, (**C**) acetic acid, (**D**) oxalic acid, and fumaric acid.

**Figure 4 foods-11-01004-f004:**
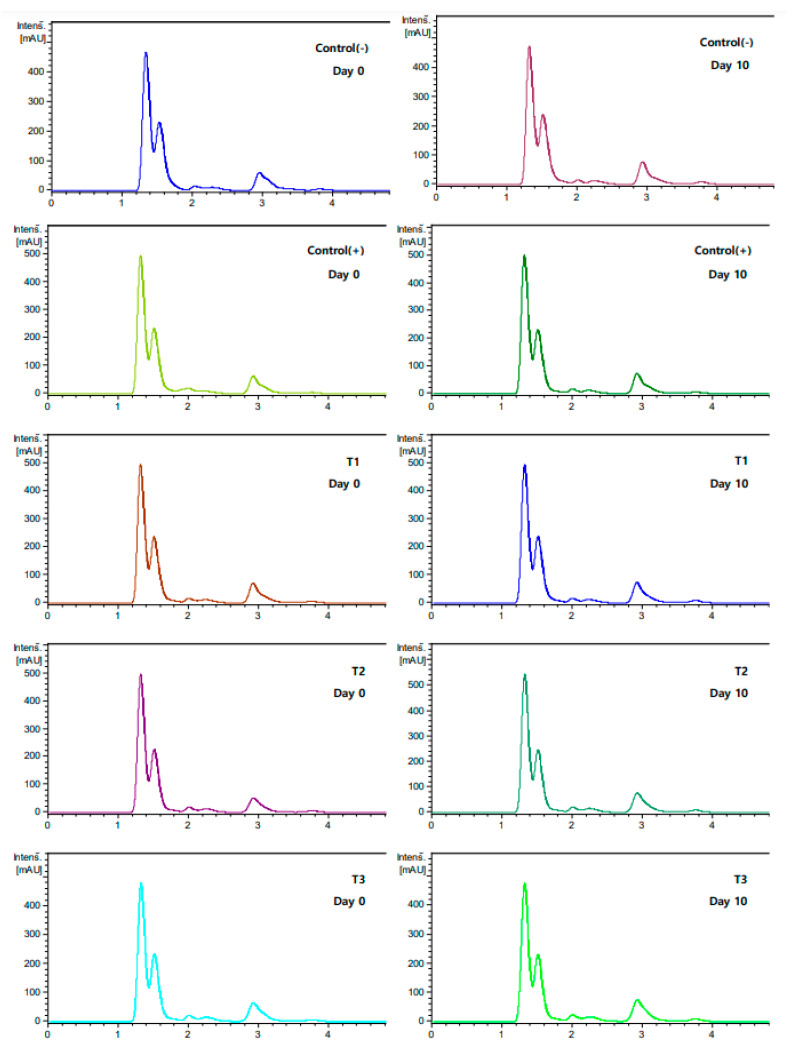
Chromatogram of the samples of the organic acids.

**Table 1 foods-11-01004-t001:** Formulation of pork patties with fermented soy sauce (%).

	Control (−)	Control (+)	T1	T2	T3
Pork	85	85	85	85	85
Ice	13.4	13.3	12.4	10.4	8.4
Salt	1.6	1.6	1.6	1.6	1.6
Ascorbic acid	-	0.1	-	-	-
Fermented soy sauce	-	-	1	3	5
Total	100	100	100	100	100

**Table 2 foods-11-01004-t002:** Physicochemical characteristics of pork patties with fermented soy sauce.

Item	Control (−)	Control-1 (+)	T1	T2	T3
Moisture (%)	73.28 ± 2.24	70.26 ± 5.66	71.16 ± 5.21	71.49 ± 3.75	71.10 ± 4.00
Protein (%)	18.34 ± 0.98	17.79 ± 1.68	17.77 ± 1.16	19.02 ± 1.49	18.21 ± 1.86
Fat (%)	5.25 ± 0.43	5.43 ± 0.34	5.32 ± 0.40	5.05 ± 0.33	5.17 ± 0.13
Ash (%)	2.63 ± 0.78 ^ab^	2.19 ± 0.75 ^b^	2.26 ± 0.64 ^b^	3.00 ± 0.47 ^a^	3.16 ± 0.39 ^a^
pH	5.94 ± 0.13 ^a^	5.79 ± 0.08 ^b^	5.96± 0.08 ^a^	5.97± 0.08 ^a^	5.97± 0.1 ^a^

^a,b^ Means ± SD with different superscripts in the same row differ significantly (*p* < 0.05).

**Table 3 foods-11-01004-t003:** WHC of pork patties with fermented soy sauce.

Item	Control (−)	Control (+)	T1	T2	T3
WHC (%)	60.81 ± 3.56 ^ab^	53.52 ± 6.91 ^b^	64.40 ± 8.89 ^a^	60.64 ± 9.09 ^ab^	61.82 ± 8.17 ^a^

^a,b^ Means ± SD with different superscripts in the same row differ significantly (*p* < 0.05).

**Table 4 foods-11-01004-t004:** Textural properties of pork patties with fermented soy sauce.

Item	Control (−)	Control (+)	T1	T2	T3
Hardness (g)	1394.89 ± 30.76 ^c^	1540.56 ± 86.94 ^b^	1542.78 ± 33.83 ^b^	1587.78 ± 49.55 ^a^	1707.89 ± 65.08 ^a^
Springiness (%)	72.95 ± 7.18	72.80 ± 4.84	74.04 ± 6.04	79.11 ± 11.15	80.30 ± 15.46
Cohesiveness (%)	56.82 ± 5.42	58.34 ± 9.22	59.64 ± 9.33	61.93 ± 6.58	58.96 ± 5.61
Chewiness (g)	576.66 ± 20.03 ^b^	647.79 ± 71.16 ^b^	672.77 ± 39.52 ^b^	789.20 ± 45.04 ^a^	810.24 ± 94.10 ^a^

^a–c^ Means ± SD with different superscripts in the same row differ significantly (*p* < 0.05).

**Table 5 foods-11-01004-t005:** Sensory evaluation of pork patties with fermented soy sauce.

Item	Control (−)	Control (+)	T1	T2	T3
Color	3.53 ± 0.83	3.73 ± 1.10	3.87 ± 0.92	4.07 ± 0.96	4.13 ± 0.92
Flavor	3.53 ± 0.83	3.47 ± 0.83	3.40 ± 0.83	3.07 ± 1.16	3.07 ± 1.03
Juiciness	3.47 ± 0.99	3.00 ± 1.13	3.13 ± 0.92	3.13 ± 1.13	3.20 ± 1.26
Saltiness	2.80 ± 1.08 ^a^	2.87 ± 1.06 ^a^	2.87 ± 0.83 ^a^	2.20 ± 0.68 ^a^	1.40 ± 0.51 ^b^
Chewiness	3.87 ± 0.92	3.47 ± 0.99	3.67 ± 0.98	3.80 ± 0.86	3.73 ± 1.03
Overall	3.53 ± 0.83 ^a^	3.20 ± 0.86 ^a^	3.27 ± 0.80 ^a^	3.00 ± 0.76 ^ab^	2.40 ± 0.91 ^b^

^a,b^ Means ± SD with different superscripts in the same row differ significantly (*p* < 0.05).

**Table 6 foods-11-01004-t006:** Meat color of pork patties with fermented soy sauce.

Item	Control (−)	Control (+)	T1	T2	T3
L	57.24 ± 5.39 ^ab^	59.62 ± 5.42 ^a^	59.00 ± 4.41 ^a^	52.39 ± 6.25 ^b^	54.82 ± 5.60 ^ab^
a	11.93 ± 1.11	10.81 ± 1.79	11.98 ± 4.41	10.97 ± 2.46	11.20 ± 0.91
b	12.24 ± 1.15	11.73 ± 1.28	12.61 ± 2.23	13.73 ± 1.42	13.42 ± 3.37

^a,b^ Means ± SD with different superscripts in the same row differ significantly (*p* < 0.05).

**Table 7 foods-11-01004-t007:** Storage properties of pork patties with fermented soy sauce.

Item		Control (−)	Control (+)	T1	T2	T3
TBA (mg malonaldehyde/1000 g)	Day 0	0.31 ± 0.12 ^b^	0.40 ± 0.35 ^ab^	0.60 ± 0.28 ^a^	0.50 ± 0.37 ^ab^	0.46 ± 0.35 ^ab^
	Day 5	1.01 ± 0.35 ^ab^	0.45 ± 0.09 ^d^	1.13 ± 0.64 ^a^	0.84 ± 0.32 ^bc^	0.65 ± 0.21 ^cd^
	Day 10	1.34 ± 0.34 ^a^	0.49 ± 0.15 ^c^	1.34 ± 0.56 ^a^	0.88 ± 0.39 ^b^	0.74 ± 0.35 ^b^
TMC (log CFU/g)	Day 0	6.32 ± 0.25 ^ab^	6.87 ± 0.73 ^a^	6.38 ± 0.23 ^ab^	6.09 ± 0.10 ^b^	6.08 ± 0.13 ^b^
	Day 5	7.79 ± 0.72 ^ab^	6.87 ± 0.63 ^b^	7.70 ± 0.67 ^ab^	8.17 ± 0.93 ^a^	7.82 ± 0.21 ^ab^
	Day 10	7.72 ± 0.83	7.46 ± 0.69	7.84 ± 0.69	7.67 ± 0.29	7.54 ± 0.48

^a–d^ Means ± SD with different superscripts in the same row differ significantly (*p* < 0.05).

**Table 8 foods-11-01004-t008:** Organic acid contents of the patty at different storage times (Day 0, Day 10).

(mg/100 g)	Control (−)		Control (+)		T1		T2			T3	
	Day 0	Day 10	Day 0	Day 10	Day 0	Day 10	Day 0	Day 10	Day 0	Day 10	Day 0
Citric acid	11.41 ± 0.05 ^cd^	13.54 ± 0.05 ^a^	9.91 ± 0.81 ^e^	12.88 ± 0.05 ^a^	11.18 ± 0.05 ^d^	12.24 ± 0.36 ^bc^	6.97 ± 0.84 ^f^	11.15 ± 0.99 ^d^	9.09 ± 0.52 ^e^	9.86 ± 0.11 ^e^	9.86 ± 0.11 ^e^
Malonic acid	1.16 ± 0.42 ^f^	2.19 ± 0.27 ^e^	2.98 ± 0.31 ^abc^	3.63 ± 0.00 ^a^	2.51 ± 0.53 ^cd^	2.70 ± 0.01 ^bcd^	3.03 ± 0.18 ^abc^	1.80 ± 0.12 ^ef^	3.38 ± 0.09 ^ab^	3.06 ± 0.87 ^abc^	3.06 ± 0.87 ^abc^
Acetic acid	3.34 ± 0.10 ^c^	4.54 ± 0.46 ^abc^	3.78 ± 0.33 ^bc^	4.60 ± 0.00 ^abc^	4.29 ± 0.97 ^abc^	5.12 ± 0.23 ^a^	4.63 ± 0.13 ^abc^	5.01 ± 0.14 ^ab^	5.06 ± 0.19 ^ab^	4.52 ± 1.88 ^abc^	4.52 ± 1.88 ^abc^
Oxalic acid	1.36 ± 0.02 ^d^	1.52 ± 0.02 ^cd^	1.50 ± 0.06 ^cd^	1.64 ± 0.06 ^bc^	1.62 ± 0.05 ^bc^	1.66 ± 0.04 ^abc^	1.87 ± 0.24 ^a^	1.81 ± 0.20 ^ab^	1.68 ± 0.10 ^abc^	1.71 ± 0.029 ^abc^	1.71 ± 0.029 ^abc^
Fumaric acid	nd	nd	0.01 ± 0.00 ^d^	0.01 ± 0.00 ^d^	0.01 ± 0.00 ^d^	0.01 ± 0.00 ^d^	0.01 ± 0.00 ^d^	0.12 ± 0.00 ^a^	0.10 ± 0.00 ^c^	0.12 ± 0.00 ^b^	0.12 ± 0.00 ^b^

^a–f^ Means ± SD with different superscripts in the same row differ significantly (*p* < 0.05).

## Data Availability

Upon reasonable request, the datasets of this study can be available from the corresponding author.

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
