# Peer review of "Effect of Addition of Fermented Soy Sauce on Quality Characteristics of Pork Patties during Refrigerated Storage"

_foods, 2022, doi:10.3390/foods11071004_

Round 1

Reviewer 1 Report

Abstract: it is not clear what is the meaning  of "tissue characteristics", "storage characteristics" (dies it mean extended shelf life?).

Introduction:

L34: main reason for meat to spoil is the high moisture content.

L47: does it mean that all the salt in soysauce is coming from the soybean and not extra salt is added in the process?

L49: originated during fermentation...

L55: are MRP already in the soy sauce when this is incorporated to the meat patties? This sentence indicates that they are only formed after cooking and in this research the soy was added to fresh pork meat.

L78: Does it mean that three independent batches were generated for each one of the 5 recipies?

L83: was it aprox. 1g or aprox 0.6-0.8g?

Line 87: why the samples, if coming from cold storage, needed to be cooled? How the samples were dried (temperature and time)

Are any of the protocols here employed for proximate composition referenced or following AOAC protocols?

L124: at what temperature?

L129: what is fat coefficient?

Section 2.4: this is the first time I see a rheometer used to determine shear force on patties. I feel this is not the right method. 

2.5: were this panel trained, how they were trained if so? How the samples were prepared for the test? Raw, cook....?

2.9: what were the factors used for this statistical analysis? Were the different batches considered as a random or fixed effects?

L234: the authors do not consider that salt is just coming from the soy sauce addition. No fermentation should happen during manufacturing.

L249: isoelectric

L250: pH

Table 4: hardness units are in Newtons of Kg, not in %, same for some other parameters in this table. Also, SD is huge, for some treatments (higher than 20%) so hard to get conclusions.

L283: same salt was added as table salt, but soy sauce has a significant amount of salt dissolved, and it is hard to believe that the highest addition of soy sauce has the lowest saltiness. Also, it seems that higher soy sauce lead to lower overall score, this has not been discussed.

L314: authors said that 1% has no effect, but initial TBAR value was very high. Also, i don't see that 3% or 5% is simialr to ascorbic acid. Even the statistic results says are different.

TMC are not consistent, some cases it increases from 5 to 10 days, in other decreases.... why is this?

Figure 3 rather than standards, it would be more relevant to plot an actual chromatogram from the samples.

3.7: why this acid were monitored? What is the meaning of their presence in the patties? Is not lactic acid more relevant?

Line 396: still don;t know what tissue caracteristics means

Based on this experimental design is not possible to conclude if the effect is coming from the soy sauce or due to the extra salt amount in the formulation coming from the soy sauce. What about health implications associated to higher sodium intake?

Author Response

Thank you for reviewing my paper. Please see the attachment.

Reviewer 2 Report

line 32 is it really salt or iron can be called nutrients? I think it should be replaced with another term. The nutrients are protein, carbohydrates and fats. 

line 34 "Lipid oxidation depends on the meat's chemical composition, light, oxygen," if it should be written on availability, reactivity, oxygen and light levels 

line 42 I think nettle is a herbal material, not a vegetable .. just like curry

line 49-50 .. I don't know if it makes sense to repeat the ingredients, you can refer to those mentioned in the previous sentence  

line 63-65- should be graphically (paragraph) separated as the purpose of the work 

line 83- it makes no sense to say that the sample was 1 g (0.6-0.8 g). Please write that samples of about 1g were taken and taking into account their actual mass - the moisture content was converted to 1g 

linia 102- co znaczy"dry matter"? If it was really dried, then how under what conditions, because it is important for changes 

general note on the materials and methods chapter - no information about the number of repetitions when performing all the analysis. I think in the abstract there was information that the experiment was performed 3 times, but this does not mean exactly the same as the execution in three repetitions 

If it was actually dried, how in Table 2 and 4,5,6- too, why SD was determined only for ash and pH

Author Response

제 논문을 검토해주셔서 감사합니다. 첨부파일을 참조하시기 바랍니다.

Reviewer 3 Report

Research topic is sufficiently justified. Some errors in Materials and methods need to be corrected and results are needed to be discussed in more detail.

Abstract:

Line 12: Since sample with no additives is termed as “control”, use different notation for sample with 0.1% ascorbic acid instead of using “control 1”

Introduction:

Line 45: After reference 10, reference 11 should come in sequence. Correct throughout accordingly.

Materials and Methods

Table 1: What is the concentration of ascorbic acid?

Line 82: Write “Moisture Content”

Determination of Moisture Content: Processing is not explained correctly. Nothing about drying of sample is mentioned. If any standard method is used, author can simply mention reference of the method and formula.

Determination of crude ash: Mention the reference of the method. If any standard method is used, author can simply mention reference of the method and formula. Formula should be written by using “insert formula” option.

Determination of crude fat: Add the reference of the method and write formula using “insert formula” option.

Line 117: “All samples were homogenized at 550 × g for 30 s”, meaning of “550 x g” is not clear.

Line 126: “centrifuged for 10 min at 45 × g.” Remove “x”. Use insert formula option for writing formula.

Line 132: “Texture properties analysis” write “Texture Profile Analysis”

Line 142: Instead of “overall preference”, use term “overall acceptance”. Change throughout the manuscript.

Determination of TMC & TBA: Write reference

Results and Discussion

Physicochemical Properties: It is observed that, protein and fat content of the sample initially decrease and then increases with increasing concentration of fermentation soya sauce. The reason for this trend is not clear.

Line 250: Replace “Ph” with “pH”

Textural Properties: Only hardness is discussed. Effect on springiness, cohesiveness and chewiness need to be discussed in detail.

Sensory Analysis: All of the parameters are not sufficiently discussed. Overall acceptability needs to be discussed in detail to analyze which combination is accepted by consumers.

Line 306: Remove word “ground”

General Comment: Language need to be improved and grammatical errors need to be corrected.

Author Response

(The authors gave the same response as above.)

Round 2

Reviewer 1 Report

IF the panel was trained, or followed particular guidelines, it needs to be explained in the methodology. As well as the demographics (gender, age, etc). Also the cooking protocol needs to be described (core temperature, way of cooking, etc)

L301: still is confusing how saltiness is explained, if a lower score is higher saltines it should be clearly explained. AS it is, leads to confusion.

L349: there was not significant differences re TMC, but when compared to control +. As salt amount is different between groups with similar initial TCM, it can't be concluded that osmotic shock is having an effect. 

Health implications re extra salt were not discussed.

L428: pH

Author Response

제 논문을 검토해주셔서 감사합니다. 첨부파일을 참조하시기 바랍니다.

Reviewer 3 Report

The authors has revised the article and I am pleased to advise publication of the article.

Author Response

(The authors gave the same response as above.)
